# Risk factors and predictive model construction for lower extremity arterial disease in diabetic patients

Yingjie Kuang[1], Zhixin Cheng[2], Jun Zhang[1], Chunxu Yang[1], Yue Zhang[2]*

**1** First School of Clinical Medicine, Shandong University of Traditional Chinese Medicine, Jinan, China,
**2** Department of Peripheral Vascular Disease, Affiliated Hospital of Shandong University of Traditional Chinese Medicine, Jinan, China

* zhangyue771@163.com

## Abstract

### Objective

To understand the prevalence and associated risk factors of lower extremity arterial disease (LEAD) in Chinese diabetic patients and to construct a risk prediction model.

### Methods

Data from the Diabetes Complications Warning Dataset of the China National Population Health Science Data Center were used. Logistic regression analysis was employed to identify related factors, and machine learning algorithms were used to construct the risk prediction model.

### Results

The study population consisted of 3,000 patients, with 476 (15.9%) having LEAD. Multivariate regression analysis indicated that male gender, atherosclerosis, carotid artery stenosis, fatty liver, hematologic diseases, endocrine disorders, and elevated glycosylated serum proteins were independent risk factors for LEAD. The risk prediction models constructed using Logistic regression and MLP algorithms achieved moderate discrimination performance, with AUCs of 0.73 and 0.72, respectively.

### Conclusion

Our study identified the risk factors for LEAD in Chinese diabetic patients, and the constructed risk prediction model can aid in the diagnosis of LEAD.

## Introduction

Lower extremity arterial disease (LEAD) is significantly associated with diabetes, with the risk increasing as the severity of diabetes worsens [1]. By 2035, the number of diabetes patients is

**Data Availability Statement:** The data that support the findings of this study are openly available in the Diabetes Complications Warning Dataset at https://www.ncmi.cn/phda/dataDetails.do?id=CSTR:A0006.11.A0005.201905.000282-V1.0.

**Funding:** This work was supported by the Shandong Provincial Natural Science Foundation (ZR2022MH268). The funders had no role in study design, data collection and analysis, decision to publish, or preparation of the manuscript.

expected to increase rapidly to approximately 600 million people [2]. Diabetic patients are at a higher risk of developing LEAD compared to non-diabetic patients [3, 4]. Consequently, the prevalence of lower extremity arterial disease in diabetic patients (LEADDP) is expected to increase significantly in the near future. However, statistics indicate that over 60% of LEAD patients may be asymptomatic, leading to early stages of the disease often being overlooked [5]. Compared to non-diabetic LEAD patients, diabetic peripheral neuropathy may mask LEAD symptoms, making the diagnosis of LEADDP more challenging. Current guidelines recommend initial screening for LEAD based on patient interviews and clinical examinations, using the Fontaine or Rutherford scales for assessment [6, 7]. This diagnosis heavily relies on the expertise of specialists and examination conditions, and the diagnostic performance is not ideal [8].

LEAD increases the risk of foot ulcers and lower limb amputations, significantly impacting the quality of life and treatment for diabetic patients, while also imposing a substantial economic burden on society [9]. Therefore, identifying risk factors and accurately predicting LEADDP risk is crucial for the early initiation of prevention and treatment in high-risk patients. In recent years, constructing risk prediction models using machine learning algorithms has become increasingly popular in medical research. These algorithms can automatically select informational variables and capture nonlinear relationships between variables, enhancing predictive capabilities.

The primary aim of this study is to describe the prevalence of LEAD in diabetic patients and to identify potential risk factors. Subsequently, we trained machine learning models to predict LEADDP risk by integrating all available clinical objective information.

## Data and methods

In this study, we used the Diabetes Complications Warning Dataset obtained from the China National Population Health Science Data Center to analyze the significant risk factors for LEADDP and to construct a risk prediction model. This dataset includes general information, vital signs, laboratory measurements, and comorbidities for 3,000 diabetic patients, totaling 81 variables (S1 Table). It encompasses traditional risk factors [1, 8]: dyslipidemia, hypertension, cardiovascular diseases, kidney disease, inflammation, obesity, male gender, aging, and infections. Due to the complex pathogenesis of LEADDP and the unclear associated risk factors, we also included more information on comorbidities and additional laboratory indicators in order to identify new risk factors and avoid omissions. We aimed to construct a risk prediction model that can be used by non-specialists; hence, treatment factors were not included.

In this study, we utilized traditional statistical methods to describe the cohort's demographic and health characteristics. For normally distributed continuous variables, independent sample t-tests or one-way ANOVA were employed to compare the means ± standard deviations between groups. Categorical variables were presented as frequencies (percentages), and the chi-square test was used to assess differences between groups. LEADDP was taken as the dependent variable, while general information, vital signs, laboratory measurements, and comorbidities were considered independent variables to explore the influence of various risk factors. Based on the results of the univariate analysis, factors with a p-value less than 0.1 were included in the multivariate analysis. We examined the correlations between all variables and visualized data with strong associations (Fig 1). Before training the model, we excluded incomplete cases from the dataset, resulting in the removal of 15 cases. After this preprocessing step, a total of 2,985 cases were included in the model. The collinearity of the included factors was examined. The dataset was randomly split into training and validation sets in a 7:3 ratio, with LEADDP as the dependent variable. The training set included 2,089 cases, with 296 cases of

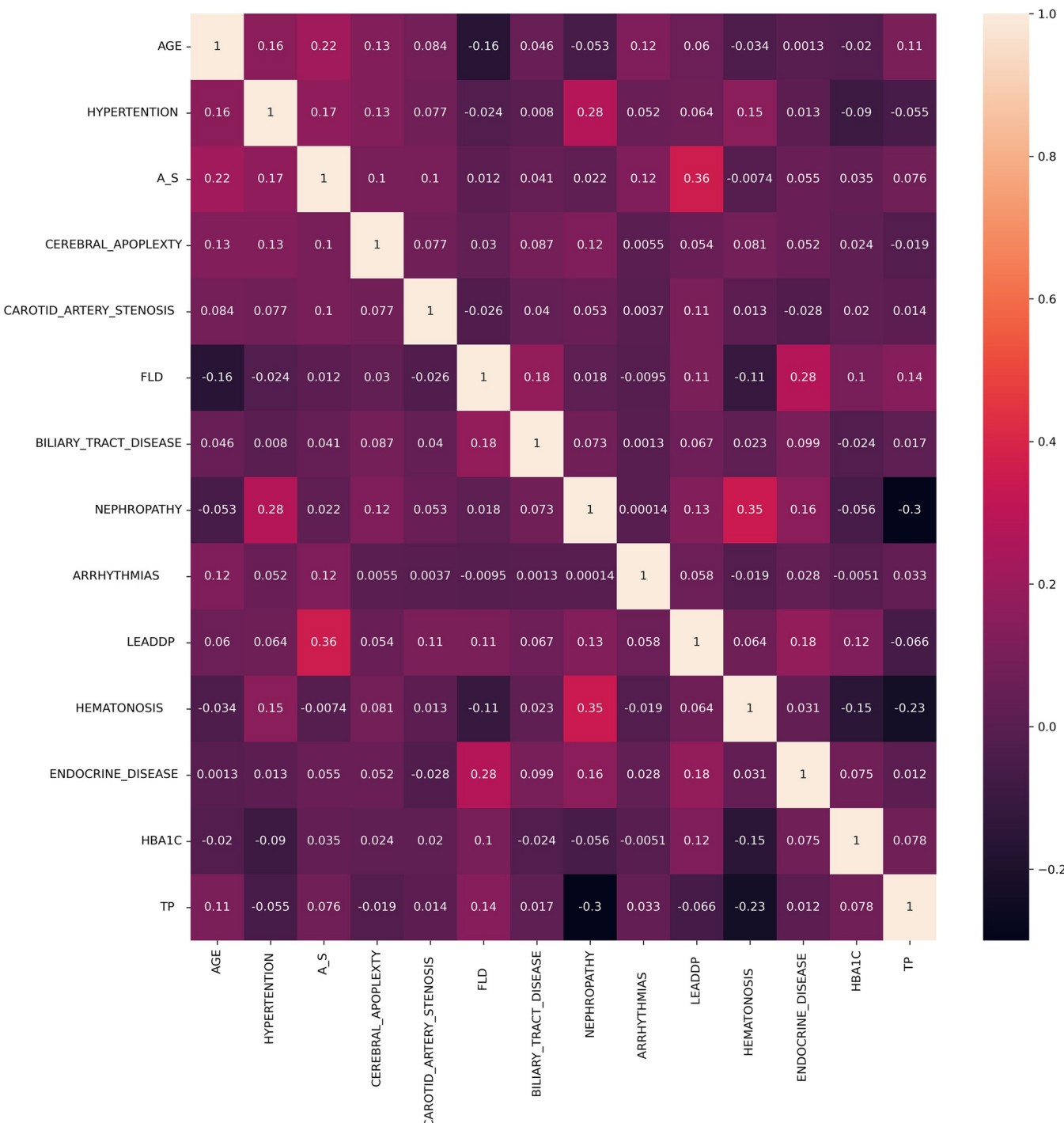

**Fig 1. Heatmap of variable correlations.** Dark blue areas indicate strong negative correlations, while dark red areas indicate strong positive correlations. Light colors show weaker correlations. Abbreviations: AS, Atherosclerosis; FLD, Fatty Liver Disease; HBA1C, Hemoglobin A1c; TP, Total Protein.

LEADDP and 1,793 cases of non-LEADDP. The validation set included 896 cases, with 127 cases of LEADDP and 769 cases of non-LEADDP (Table 1). We used the training set data to construct the LEADDP risk prediction model and validated it using the validation set data.

**Table 1. Characteristics of samples in the training and validation sets.**

|  | NO LEADDP | LEADDP | Total |
|---|---|---|---|
| Training set | 1793 | 296 | 2089 |
| Validation set | 769 | 127 | 896 |
| Total | 2562 | 423 | 2985 |

## Results

Among the 3,000 diabetic patients included in this study, 476 (15.9%) had LEADDP. The average age was 57.8 years, with 37.5% being female. Significant differences were observed between patients who had LEADDP and those who did not in terms of general information, vital signs, laboratory tests, and comorbidities (Table 2).

LEADDP patients were predominantly male, older in age, and had higher proportions of hypertension, stroke, carotid artery stenosis, fatty liver, biliary diseases, kidney disease, arrhythmias, hematologic diseases, and other endocrine disorders. They also exhibited poorer glycemic control, with the majority having atherosclerosis in other areas. Additionally, LEADDP patients had higher levels of low-density lipoprotein and urinary creatinine, and lower levels of urinary microalbumin, total protein, serum albumin, aspartate aminotransferase, C-reactive protein, and globulin. Other laboratory indicators showed no statistically significant differences. In the multivariable analysis, all variables that showed a strong association with LEADDP in the univariate analysis ($P < 0.1$) were included. The results indicated that gender, atherosclerosis, carotid artery stenosis, fatty liver, hematologic diseases, endocrine disorders, and glycosylated serum protein were significantly associated with LEADDP (Table 3).

We constructed five LEADDP risk prediction models using Python, specifically the Logistic regression, decision tree, random forest, k-nearest neighbors, and neural network algorithms. We plotted the ROC curves for each algorithm (Fig 2). Among them, the risk prediction models based on the Logistic regression and MLP algorithms achieved the best performance, with AUCs of 0.73 and 0.72, respectively.

## Discussion

In our study, approximately 15.9% of Chinese diabetic patients had LEAD. Male gender, atherosclerosis, carotid artery stenosis, fatty liver, hematologic diseases, other endocrine disorders, and glycated serum protein were found to be independently associated with the prevalence of LEADDP. The risk prediction model constructed using Logistic regression and MLP algorithms achieved the best performance, with moderate discriminative ability.

In our study, approximately 15.9% of Chinese diabetic patients had LEADDP, whereas another study from the southern coastal region of China reported a prevalence of about 4.9% [10]. The reason for this difference is not yet clear but may be related to regional variations. Our study data are from a medical center in northern China, where the climate is colder compared to the south. The diagnosis of LEAD primarily relies on ABI (Ankle-Brachial Index). At low temperatures, the pressure in the distal arteries of the lower limbs significantly decreases, which reduces the ABI value [11]. Therefore, we believe this difference may be due to the influence of different climates in the regions where the studies were conducted. Additionally, there are significant differences in dietary habits, lifestyles, and economic levels among people in different provinces of China, which may, to some extent, affect the development of diabetes and its complications. Existing research data indicate that the prevalence of diabetes varies across different provinces in China [12]. However, there have been no epidemiological reports specifically on LEADDP in China.

**Table 2. Baseline characteristics according to the presence of LEADDP.**

| Category/variable | | NO LEADDP | LEADDP | *P* value |
|---|---|---|---|---|
| Sex, n (%) | male | 1560 (61.8) | 314 (66) | 0.089 |
| | female | 964 (38.2) | 162 (34) | |
| Age, years | | 57.51±11.28 | 59.3±10.37 | **0.001** |
| Height, meter | | 166.51±7.03 | 166.34±8.54 | 0.665 |
| Weight, kg | | 73±13.00 | 73.56±12.43 | 0.392 |
| SBP, mmHg | | 138.22±20.84 | 141.18±21.72 | **0.005** |
| DBP, mmHg | | 80.40±12.01 | 80.82±11.57 | 0.48 |
| Heart Rate, n/min | | 79.88±26.81 | 69.64±35.03 | 0.107 |
| BMI, kg/m$^2$ | | 26.25±3.82 | 26.58±3.61 | 0.088 |
| Hypertention, n (%) | no | 835(33.1) | 119(25) | **0.001** |
| | yes | 1689(66.9) | 357(75) | |
| Hyperlipidemia, n (%) | no | 1973(78.3) | 371(77.9) | 0.952 |
| | yes | 551(21.8) | 105(22.1) | |
| AS, n (%) | no | 1420(56.3) | 37(7.8) | **<0.0001** |
| | yes | 1104(43.7) | 439(92.2) | |
| Cerebral Apoplexty, n (%) | no | 2351(93.1) | 425(89.3) | **0.004** |
| | yes | 173(6.9) | 51(10.7) | |
| Carotid Artery Stenosis, n (%) | no | 2440(96.7) | 431(90.5) | **<0.0001** |
| | yes | 84(3.3) | 45(9.5) | |
| FLD, n (%) | no | 1793(71) | 270(56.7) | **<0.0001** |
| | yes | 731(29) | 206(43.3) | |
| Cirrhosis, n (%) | no | 2483(98.4) | 470(98.7) | 0.689 |
| | yes | 41(1.6) | 6(1.3) | |
| CLD, n (%) | no | 2191(968) | 400(84) | 0.109 |
| | yes | 333(13.2) | 76(16) | |
| Pancreatic Disease, n (%) | no | 2482(98.3) | 470(98.7) | 0.563 |
| | yes | 42(1.7) | 6(1.3) | |
| Biliary Tract Disease, n (%) | no | 2190(86.8) | 383(80.5) | **<0.0001** |
| | yes | 334(13.2) | 93(19.5) | |
| Nephropathy, n (%) | no | 1520(60.2) | 203(42.6) | **<0.0001** |
| | yes | 1004(39.8) | 273(57.4) | |
| Renal Faliure, n (%) | no | 2375(94.1) | 442(92.9) | 0.347 |
| | yes | 149(5.9) | 34(7.1) | |
| Nervous System Disease, n (%) | no | 2382(94.4) | 44(92.9) | 0.202 |
| | yes | 142(5.6) | 34(7.1) | |
| CHD, n (%) | no | 1693(67.1) | 322(67.6) | 0.832 |
| | yes | 831(32.9) | 154(32.4) | |
| MI, n (%) | no | 2363(93.6) | 447(93.9) | 0.839 |
| | yes | 161(6.4) | 29(6.1) | |
| CHF, n (%) | no | 2351(93.1) | 437(91.8) | 0.329 |
| | yes | 173(6.9) | 39(8.2) | |
| Arrhythmias, n (%) | no | 2392(94.8) | 434(91.2) | **0.003** |
| | yes | 132(5.2) | 42(8.8) | |
| Respiratory System Disease, n (%) | no | 2121(84) | 407(85.5) | 0.451 |
| | yes | 403(16) | 69(14.5) | |
| Hematonosis, n (%) | no | 2175(86.2) | 381(80) | **0.001** |
| | yes | 349(13.8) | 95(20) | |

(*Continued*)

**Table 2.** (Continued)

| Category/variable | | NO LEADDP | LEADDP | *P* value |
|---|---|---|---|---|
| Rheumatic Immunity, n (%) | no | 2432(96.4) | 465(97.7) | 0.169 |
| | yes | 92(3.6) | 11(2.3) | |
| Pregnant, n (%) | no | 2515(99.6) | 475(99.8) | 0.715 |
| | yes | 9(0.4) | 1(0.2) | |
| Endocrine Disease, n (%) | no | 1773(70.2) | 225(47.3) | **<0.0001** |
| | yes | 751(29.8) | 251(52.7) | |
| MEN, n (%) | no | 2430(96.3) | 462(97.1) | 0.426 |
| | yes | 94(3.7) | 14(2.9) | |
| PCOS, n (%) | no | 2521(99.9) | 476(100) | 1 |
| | yes | 3(0.1) | 0(0) | |
| Digestive Carcinoma, n (%) | no | 2385(94.5) | 462(97.1) | 0.022 |
| | yes | 139(5.5) | 14(2.9) | |
| Urologic Neoplasms, n (%) | no | 2496(98.9) | 473(99.4) | 0.462 |
| | yes | 28(1.1) | 3(0.6) | |
| Gynecolgical Tumor, n (%) | no | 2434(96.4) | 467(98.1) | 0.068 |
| | yes | 90(3.6) | 9(1.9) | |
| Breast Tumor, n (%) | no | 2515(99.6) | 475(99.8) | 0.715 |
| | yes | 9(0.4) | 1(0.2) | |
| Lung Tumor, n (%) | no | 2478(98.2) | 467(98.1) | 1 |
| | yes | 46(1.8) | 9(1.9) | |
| Intracranial Tumor, n (%) | no | 2512(99.5) | 472(99.2) | 0.493 |
| | yes | 12(0.5) | 4(0.8) | |
| Other Tumor, n (%) | no | 2312(91.6) | 442(92.9) | 0.366 |
| | yes | 212(8.4) | 34(7.1) | |
| GLU, mmol/L | | 8.40±3.93 | 8.69±3.72 | 0.139 |
| GLU_2H, mmol/L | | 14.52±4.58 | 15.81±4.46 | **0.001** |
| HBA1C, % | | 7.70±1.71 | 8.29±1.79 | **<0.0001** |
| GSP, μmol/L | | 224.11±79.15 | 237.69±79.13 | **0.005** |
| TG, mmol/L | | 2.03±1.65 | 1.99±1.50 | 0.704 |
| TC, mmol/L | | 4.60±1.38 | 4.69±1.61 | 0.233 |
| HDL_C, mmol/L | | 1.07±0.32 | 1.06±0.30 | 0.762 |
| LDL_C, mmol/L | | 2.84±1.09 | 2.95±1.38 | 0.048 |
| FBG, g/L | | 7.73±37.90 | 8.69±42.44 | 0.621 |
| UPR_24, g/24h | | 1.44±1.50 | 1.08±1.41 | **<0.0001** |
| BU, mmol/L | | 7.07±4.97 | 7.37±4.91 | 0.222 |
| SCR, μmol/L | | 106.24±119.97 | 108.29±114.39 | 0.731 |
| UCR, mmol/24h | | 5.49±2.66 | 6.61±12.16 | 0.021 |
| SUA, μmol/L | | 328.86±103.68 | 327.81±93.56 | 0.838 |
| HB, g/L | | 131.77±23.35 | 131.4±21.94 | 0.75 |
| CP, nmol/L | | 2.21±1.53 | 2.17±2.1 | 0.777 |
| INS, μU/ml | | 14.71±39.04 | 11.61±13.54 | 0.213 |
| PCV | | 0.39±0.06 | 0.38±0.06 | 0.493 |
| PLT, $10^9$/L | | 219.18±73.16 | 213.34±63.99 | 0.105 |
| ESR, mm/h | | 27.08±29.49 | 23.3±25.81 | 0.056 |
| TBILI, μmol/L | | 11.02±12.35 | 10.79±7.14 | 0.698 |
| DBILI, μmol/L | | 3.64±9.67 | 2.99±2.16 | 0.147 |
| TP, g/L | | 65.7±7.43 | 64.36±7.51 | **<0.0001** |

(*Continued*)

**Table 2.** (Continued)

| Category/variable | NO LEADDP | LEADDP | *P* value |
|---|---|---|---|
| ALB, g/L | 39.53±5.832 | 38.67±6.1 | **0.004** |
| LDH_L, U/L | 174.29±64.78 | 171.48±54.01 | 0.377 |
| ALT, U/L | 24.47±29.252 | 21.83±15.86 | 0.055 |
| AST, U/L | 20.74±24.03 | 18.26±10.06 | **0.028** |
| GGT, U/L | 43.3±73.3 | 38.844±53.28 | 0.209 |
| ALP, U/L | 75.72±47.7 | 72.31±30.28 | 0.138 |
| LP_A, mg/dl | 30.81±43.58 | 36.84±53.03 | 0.661 |
| PL, mmol/L | 2.46±0.60 | 2.46±0.69 | 0.972 |
| PT, s | 13.15±1.28 | 13.1±1.19 | 0.413 |
| PTA | 99.19±18.88 | 98.37±19.07 | 0.387 |
| APTT, s | 36.74±7.33 | 36.42±5.61 | 0.361 |
| FIBRIN, g/L | 13.96±25.29 | 16.12±30.71 | 0.833 |
| ALB_CR, mg/g | 149.66±230.06 | 154.12±260.96 | 0.748 |
| LPS, U/L | 164.58±425.35 | 157.88±300.38 | 0.808 |
| CA199, U/L | 30.7±226.85 | 18.21±16.1 | 0.247 |
| CRP, mg/L | 1.26±2.88 | 0.85±2.15 | **0.008** |
| TH2, ng/L | 0.45±0.1 | 0.35±0.11 | 0.155 |
| IBILI, μmol/L | 7.37±4.35 | 7.78±5.59 | 0.072 |
| GLO, g/L | 26.18±4.95 | 25.68±4.56 | **0.044** |

Note: Data were expressed as mean ± SD or n (%).

Abbreviations: SBP, Systolic Blood Pressure; DBP, Diastolic Blood Pressure; BMI, Body Mass Index; CHD, Coronary Heart Disease; MI, Myocardial Infarction; CHF, Congestive Heart Failure; AS, Atherosclerosis; FLD, Fatty Liver Disease; CLD, Chronic Liver Disease; PCOS, Polycystic Ovary Syndrome; MEN, Multiple Endocrine Neoplasia; GLU, Glucose; GLU_2H, 2-hour Postprandial Glucose; HBA1C, Hemoglobin A1c; GSP, Glycated Serum Protein; TG, Triglycerides; TC, Total Cholesterol; HDL_C, High-Density Lipoprotein Cholesterol; LDL_C, Low-Density Lipoprotein Cholesterol; FBG, Fibrinogen; UPR_24, 24-hour Urinary Protein; BUN, Blood Urea Nitrogen; BU, Blood Urea; SCR, Serum Creatinine; UCR, Urine Creatinine; SUA, Serum Uric Acid; HB, Hemoglobin; CP, C-Peptide; INS, Insulin; PCV, Packed Cell Volume; PLT, Platelets; ESR, Erythrocyte Sedimentation Rate; TBILI, Total Bilirubin; DBILI, Direct Bilirubin; TP, Total Protein; ALB, Albumin; LDH_L, Lactate Dehydrogenase; ALT, Alanine Aminotransferase; AST, Aspartate Aminotransferase; GGT, Gamma-Glutamyl Transferase; ALP, Alkaline Phosphatase; LP_A, Lipoprotein(a); PL, Phospholipids; PT, Prothrombin Time; PTA, Prothrombin Activity; APTT, Activated Partial Thromboplastin Time; FIBRIN, Fibrinogen; ALB_CR, Albumin/Creatinine Ratio; LPS, Lipase; CA199, Cancer Antigen 19–9; CRP, C-Reactive Protein; TH2, T Helper Cell 2; IBILI, Indirect Bilirubin; GLO, Globulin.

The risk of LEAD increases with age, a finding also supported by our study [8]. However, existing research indicates that the number of peripheral arterial disease (PAD) cases is higher in females than in males across all age groups [13]. Since females have lower ABI values than

**Table 3. Results of the multivariate analysis.**

| | *P* value | OR (95% CI) |
|---|---|---|
| Sex | 0.018 | 1.422 (1.062, 1.904) |
| AS | <0.001 | 19.911 (12.349, 32.103) |
| Carotid Artery Stenosis | 0.002 | 2.198 (1.345, 3.591) |
| FLD | 0.004 | 1.518 (1.140, 2.023) |
| Hematonosis | <0.001 | 2.094 (1.467, 2.989) |
| Endocrine Disease | <0.001 | 2.155 (1.623, 2.862) |
| GSP | 0.014 | 1.002 (1.000, 1.004) |

Abbreviations: AS, Atherosclerosis; FLD, Fatty Liver Disease; GSP, Glycated Serum Protein.

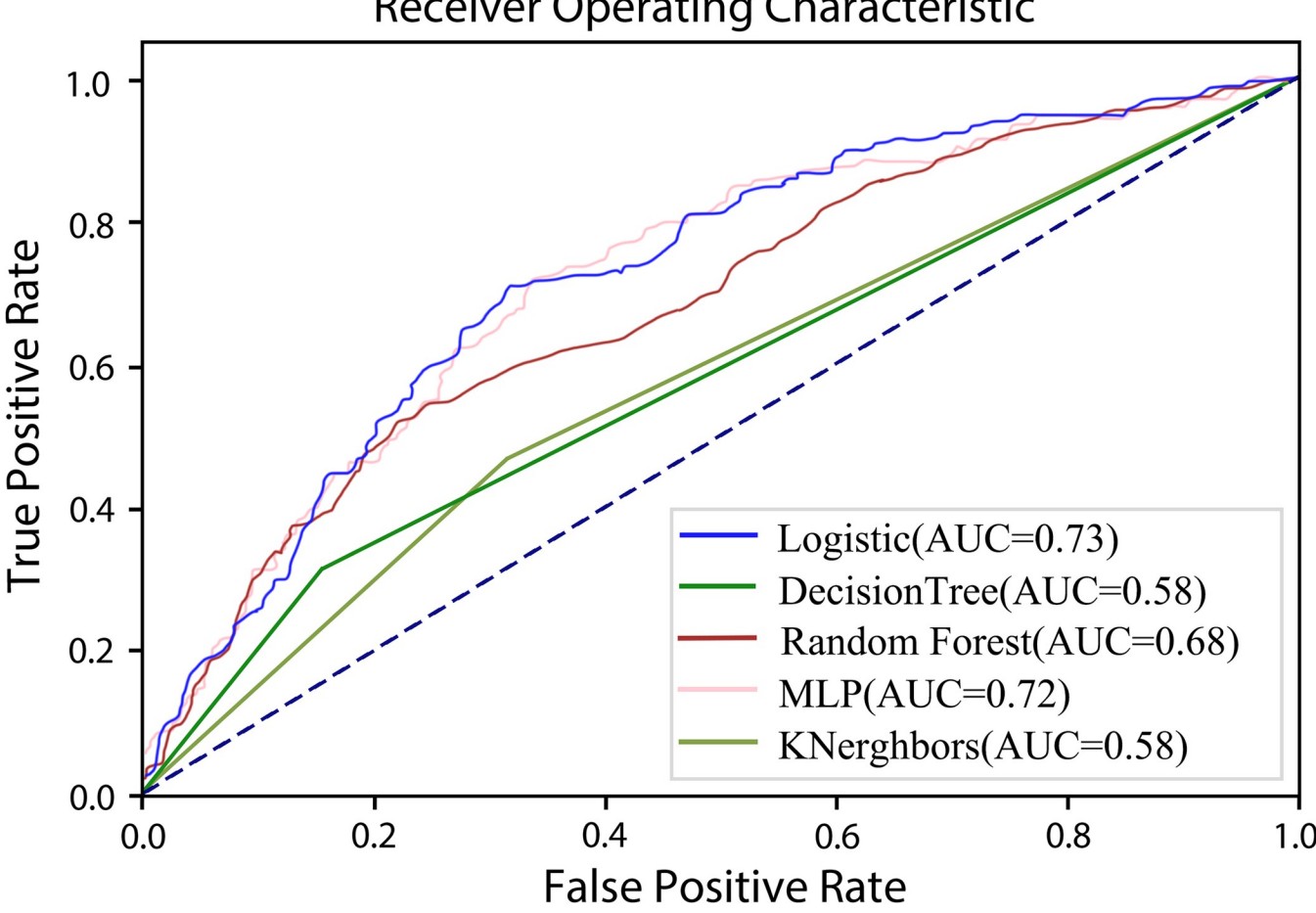

**Fig 2. AUC curves of various risk prediction models.**

males, the actual prevalence of PAD in females might be higher than current estimates [14, 15]. However, in our study, we found that there were more men than women among diabetic patients with concurrent LEAD. A study on LEAD in the diabetic population also reached similar conclusions [16]. Males have a higher risk of LEADDP compared to females, with male diabetic patients being about 1.4 times more likely to develop LEADDP than female diabetic patients. This may be related to the unique pathophysiological processes in diabetic patients.

Diabetic vascular complications share common features, with their main pathological manifestations being endothelial dysfunction and atherosclerosis [17, 18]. The relationship between diabetic macrovascular complications and LEADDP is evident and confirmed in our study. Additionally, we found that carotid artery stenosis can be considered a risk factor for LEADDP. The risk of LEADDP in patients with carotid artery stenosis is 2.198 times higher than in those without. Because symptoms are more evident, diabetic patients with carotid artery stenosis are more likely to seek medical help. Therefore, emphasizing the predictive value of carotid artery stenosis will aid in the diagnosis and early treatment of LEADDP. Due to the widespread vascular complications of diabetes, cardiovascular disease cannot be ignored [17]. However, in our study, there were no significant differences in the prevalence of complications such as coronary heart disease, myocardial infarction, and chronic heart failure. Our study shows that hypertension is associated with LEADDP, but not all types of hypertension.

Consistent with some studies, increased systolic pressure is associated with LEADDP, while differences in diastolic pressure are not statistically significant [19, 20]. This suggests that we should pay attention to the LEAD risk in diabetic patients with increased pulse pressure.

Glycemic control is considered an important measure to prevent the risk of LEADDP. Hyperglycemia promotes oxidative stress, glycoxidation, and systemic inflammation, damaging the endothelial cells of the arterial wall, leading to lipid deposition and the development of atherosclerosis [21]. Some studies have shown a link between glycemic control levels and the incidence of LEAD, indicating that poor glycemic control may lead to the development of LEADDP [10, 22, 23]. Other studies, however, have found little association between blood glucose levels and LEADDP [24, 25]. The reasons for these differences are unclear, possibly involving differences in study populations or data analysis methods. In our study, LEAD patients had poorer blood glucose control compared to those without LEAD. Although fasting blood glucose levels did not show significant differences, the postprandial 2-hour blood glucose levels were significantly elevated. Some studies have suggested that insulin resistance is an important cause of LEADDP and can be harmful even with normal blood glucose levels [26, 27]. Studies have also shown that arterial vascular damage can begin before blood glucose levels increase, with elevated insulin levels potentially being a contributing factor [21]. This may indicate that poor glycemic control and LEADDP are not the cause but the result. This may suggest a deeper connection between glycemic control and the occurrence of LEADDP, warranting further research.

Lipid abnormalities play a crucial role in the development of atherosclerosis. Unlike in the past, where only low-density lipoprotein was emphasized, both low-density lipoprotein and triglyceride abnormalities should now be considered [28]. For LEADDP, a recent study has shown that plasma concentrations of HDL-C, TC/HDL-C, and apolipoproteins are associated with the incidence of LEADDP [29]. Other studies have not found a link between lipid parameters and the incidence of LEADDP [23, 30]. These inconsistencies may be due to lipid-lowering treatments in the study populations [31]. In our study, low-density lipoprotein was significantly elevated in the LEADDP group, but other lipid indicators, particularly triglycerides, showed no statistically significant differences. Elevated low-density lipoprotein levels indicate dyslipidemia. In our study, patients with both LEAD and fatty liver had an increased risk of LEAD, and the difference in BMI was not significant. In some studies, obesity is considered a risk factor for the development of LEAD [32, 33]. However, obesity does not cause LEADDP in all populations [34]. This suggests that obesity may not be a direct risk factor for LEADDP, but rather an indicator of dyslipidemia.

Impaired renal function is not a traditional risk factor for LEAD, but studies have shown that it is associated with an increased risk of atherosclerosis, cardiovascular events, and LEAD [10, 35, 36]. In our study, individuals with LEADDP exhibited a higher prevalence of kidney disease and significantly reduced kidney function compared to those without LEADDP. Univariate analysis showed that concurrent kidney disease, UPR_24, and UCR were associated with LEADDP events, but this association was excluded after adjusting for potential confounding factors.

Certain endocrine diseases other than diabetes can also increase the risk of LEADDP. Hypothyroidism is associated with an increased risk of atherosclerosis and may cause peripheral vascular constriction [37, 38]. Patients with Cushing's syndrome have elevated cortisol levels, leading to hypertension, hyperglycemia, and lipid abnormalities, increasing the risk of atherosclerosis [39]. In our study, we found that having other endocrine diseases increases the risk of LEADDP.

In this study, hematologic diseases were considered one of the risk factors for LEADDP. Possible mechanisms include coagulation disorders, abnormal platelet counts or function, and increased blood viscosity. Abnormalities in fibrinogen, thrombin formation, and fibrin

degradation occur not only in acute thrombotic complications but may also occur in stable forms of LEAD, as research suggests [40]. Platelet abnormalities and increased blood viscosity may lead to the development and exacerbation of LEAD [41]. Currently, no studies have analyzed the correlation between hematologic diseases and LEADDP. This finding opens new avenues for research into LEADDP.

Machine learning can assist clinicians in making diagnoses, thereby reducing their workload, increasing the sensitivity of disease diagnosis, avoiding missed diagnoses, and lowering healthcare costs [42]. The pathogenesis of LEADDP is complex, and current research does not clearly define its risk factors. Machine learning algorithms can learn from existing data and identify relationships between independent and dependent variables. This capability makes them well-suited for constructing risk prediction models for LEADDP [43]. Previous studies have employed machine learning algorithms to develop risk prediction models for LEAD, achieving relatively good predictive performance [44, 45]. However, these studies did not specifically focus on diabetic patients, and thus could not capture the unique characteristics of LEAD occurrence in this population. The application of machine learning algorithms in LEAD risk prediction models specifically targeting diabetic patients remains relatively limited. In our study, we aimed to develop a preliminary machine learning model to predict LEAD in diabetic patients. Our results indicated that Logistic Regression and MLP were the most effective algorithms, exhibiting the highest AUC and superior statistical performance. Generally, the AUC ranges from 0.5 to 1.0, with values between 0.5 and 0.7 indicating low discrimination ability, values between 0.7 and 0.9 indicating moderate discrimination ability, and values above 0.9 indicating high discrimination ability [46]. In our study, the accuracy of the model's predictive ability was evaluated by calculating the area under the ROC curve. The AUC values of the risk prediction models based on Logistic regression and MLP were 0.73 and 0.72, respectively, indicating that these two models achieved moderate discrimination ability. This model is built using easily collectable objective clinical data, and it can be directly integrated with data from medical systems without the need for secondary collection. This allows for improved LEAD risk prediction without increasing the workload of medical personnel.

This study identified risk factors for LEAD in the Chinese diabetic population, further explored the mechanisms related to LEADDP, and developed a relatively user-friendly risk prediction model using machine learning algorithms. The model can assist specialists in identifying high-risk LEADDP patients. The limitations of this study include the following. First, the data used in this study were obtained from the China National Population Health Science Data Center, and while the data were collected by reliable medical centers, the comorbidities in the dataset were not clearly defined, which may pose limitations for broader application. Second, for the sake of model usability, this study did not include treatment factors, so the potential effects of medications on the outcomes have not been explored. Third, as this is a cross-sectional study, the research design does not allow us to establish causal relationships between the identified risk factors. Finally, the data primarily come from northern China, which may limit its generalizability to other regions.

## Conclusion

In conclusion, our study identified several important risk factors for LEAD in the Chinese diabetic population. These findings contribute to a deeper understanding of the pathogenesis of LEADDP. The accumulation of risk factors further increases the risk of disease, highlighting the urgent need for early diagnosis and intervention. The LEADDP risk prediction models constructed using machine learning can assist in early screening, reducing adverse outcomes and long-term risks for patients.

## Supporting information

**S1 Table. Patient characteristics in the dataset.** Abbreviations: SBP, Systolic Blood Pressure; DBP, Diastolic Blood Pressure; BMI, Body Mass Index; CHD, Coronary Heart Disease; MI, Myocardial Infarction; CHF, Congestive Heart Failure; AS, Atherosclerosis; FLD, Fatty Liver Disease; CLD, Chronic Liver Disease; PCOS, Polycystic Ovary Syndrome; MEN, Multiple Endocrine Neoplasia; GLU, Glucose; GLU_2H, 2-hour Postprandial Glucose; HBA1C, Hemoglobin A1c; GSP, Glycated Serum Protein; TG, Triglycerides; TC, Total Cholesterol; HDL_C, High-Density Lipoprotein Cholesterol; LDL_C, Low-Density Lipoprotein Cholesterol; FBG, Fibrinogen; UPR_24, 24-hour Urinary Protein; BUN, Blood Urea Nitrogen; BU, Blood Urea; SCR, Serum Creatinine; UCR, Urine Creatinine; SUA, Serum Uric Acid; HB, Hemoglobin; CP, C-Peptide; INS, Insulin; PCV, Packed Cell Volume; PLT, Platelets; ESR, Erythrocyte Sedimentation Rate; TBILI, Total Bilirubin; DBILI, Direct Bilirubin; TP, Total Protein; ALB, Albumin; LDH_L, Lactate Dehydrogenase; ALT, Alanine Aminotransferase; AST, Aspartate Aminotransferase; GGT, Gamma-Glutamyl Transferase; ALP, Alkaline Phosphatase; LP_A, Lipoprotein(a); PL, Phospholipids; PT, Prothrombin Time; PTA, Prothrombin Activity; APTT, Activated Partial Thromboplastin Time; FIBRIN, Fibrinogen; ALB_CR, Albumin/Creatinine Ratio; LPS, Lipase; CA199, Cancer Antigen 19–9; CRP, C-Reactive Protein; TH2, T Helper Cell 2; IBILI, Indirect Bilirubin; GLO, Globulin.
(PDF)

## Acknowledgments

We thank the China National Population Health Science Data Center for providing data support. We also appreciate the constructive comments from the editors and each reviewer during the revision of our manuscript.

## Author Contributions

**Conceptualization:** Yingjie Kuang, Chunxu Yang.

**Formal analysis:** Yingjie Kuang.

**Methodology:** Jun Zhang.

**Supervision:** Yue Zhang.

**Validation:** Zhixin Cheng.

**Writing – original draft:** Yingjie Kuang.

**Writing – review & editing:** Yingjie Kuang, Zhixin Cheng, Jun Zhang, Chunxu Yang, Yue Zhang.

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
