## [Decision Letter · Decision Letter 0]

28 Aug 2024

PONE-D-24-26075Risk Factors and Predictive Model Construction for Lower Extremity Arterial Disease in Diabetic PatientsPLOS ONE

Dear Dr. Kuang,

Thank you for submitting your manuscript to PLOS ONE. After careful consideration, we feel that it has merit but does not fully meet PLOS ONE’s publication criteria as it currently stands. Therefore, we invite you to submit a revised version of the manuscript that addresses the points raised during the review process.

Please submit your revised manuscript within Oct 12 2024 11:59PM. If you will need more time than this to complete your revisions, please reply to this message or contact the journal office at plosone@plos.org. Please include the following items when submitting your revised manuscript:A rebuttal letter that responds to each point raised by the academic editor and reviewer(s). You should upload this letter as a separate file labeled 'Response to Reviewers'.A marked-up copy of your manuscript that highlights changes made to the original version. You should upload this as a separate file labeled 'Revised Manuscript with Track Changes'.An unmarked version of your revised paper without tracked changes. You should upload this as a separate file labeled 'Manuscript'.

We look forward to receiving your revised manuscript.

Kind regards,

Adeel Ahmad Khan

Academic Editor

PLOS ONE

Journal Requirements:

1. When submitting your revision, we need you to address these additional requirements.Please ensure that your manuscript meets PLOS ONE's style requirements, including those for file naming. The PLOS ONE style templates can be found at https://journals.plos.org/plosone/s/file?id=wjVg/PLOSOne_formatting_sample_main_body.pdf and https://journals.plos.org/plosone/s/file?id=ba62/PLOSOne_formatting_sample_title_authors_affiliations.pdf 2. Please note that PLOS ONE has specific guidelines on code sharing for submissions in which author-generated code underpins the findings in the manuscript. In these cases, we expect all author-generated code to be made available without restrictions upon publication of the work. Please review our guidelines at https://journals.plos.org/plosone/s/materials-and-software-sharing#loc-sharing-code and ensure that your code is shared in a way that follows best practice and facilitates reproducibility and reuse. 3. Thank you for stating the following financial disclosure: "This work was supported by the Shandong Provincial Natural Science Foundation (ZR2022MH268)."Please state what role the funders took in the study.  If the funders had no role, please state: "The funders had no role in study design, data collection and analysis, decision to publish, or preparation of the manuscript." If this statement is not correct you must amend it as needed. Please include this amended Role of Funder statement in your cover letter; we will change the online submission form on your behalf. 4. For studies involving third-party data, we encourage authors to share any data specific to their analyses that they can legally distribute. PLOS recognizes, however, that authors may be using third-party data they do not have the rights to share. When third-party data cannot be publicly shared, authors must provide all information necessary for interested researchers to apply to gain access to the data. (https://journals.plos.org/plosone/s/data-availability#loc-acceptable-data-access-restrictions)  For any third-party data that the authors cannot legally distribute, they should include the following information in their Data Availability Statement upon submission:1) A description of the data set and the third-party source2) If applicable, verification of permission to use the data set3) Confirmation of whether the authors received any special privileges in accessing the data that other researchers would not have4) All necessary contact information others would need to apply to gain access to the data.

Reviewers' comments:

Reviewer's Responses to Questions

**Comments to the Author**

1. Is the manuscript technically sound, and do the data support the conclusions?

Reviewer #1: Yes

Reviewer #2: Yes

Reviewer #3: Partly

Reviewer #4: Yes

2. Has the statistical analysis been performed appropriately and rigorously? 

Reviewer #1: Yes

Reviewer #2: No

Reviewer #3: Yes

Reviewer #4: Yes

3. Have the authors made all data underlying the findings in their manuscript fully available?

Reviewer #1: Yes

Reviewer #2: No

Reviewer #3: Yes

Reviewer #4: Yes

4. Is the manuscript presented in an intelligible fashion and written in standard English?

Reviewer #1: Yes

Reviewer #2: Yes

Reviewer #3: No

Reviewer #4: Yes

5. Review Comments to the Author

Reviewer #1: The authors analyzed 3000 diabetic patients from the Diabetes Complications Warning Dataset of the China National Population Health Science Data Center to identify risk factors of LEAD and construct its risk prediction model. With 476 patients who developed LEAD, they identified several risk factors and their logistic regression-based prediction model has moderate discrimination performance with AUCs of 0.73 and 0.72.

1. Treatment factors were excluded in the prediction model. Please clarify what treatment and also provide more detail about the rational for this decision.

2. Authors only said “missing values were handled” but no further information. How were they handled should be provided.

3. Also, testing collinearity is good. But what authors deal with collinear variables should be provided.

4. All significant predictors from univariate analysis were included in multivariate analysis. How about those non-significant in univariate analysis? If not included, will you be concerned about the potential confounding?

5. Please present the sample characteristic comparison between the training and testing sample

Reviewer #2: It needs a minor revision , of the statistical methods with better description of the patients and methods.

The results are not well detailed and needs more refinement.

The discussions are weakly describing the current literature.

Reviewer #3: The authors used data from the Diabetes Complications Warning Dataset of the China National Population Health Science Data Center and Logistic regression, decision tree, random forest, k-nearest neighbors, and neural network algorithms to test the accuracy of LEADDP risk prediction models. Several suggestions were listed as follows.

1. Please make a brief introduction and clarification of the dataset and study design the authors used. It seems to be a cross-sectional design. If so, it is inappropriate to call it incidence.

2. Table 1 should be included into supplementary.

3. The figures are very vague in the combined PDF. After downloading the original figures, it's hard to see the words alongside the bars. Please add figure legend.

4. BP high and BP low, should be changed into systolic blood pressure and diastolic blood pressure.

5. No unit is indicated in Table 2.

6. Why listing so many variables in the table 2? Are these variables all associated with LEAD? It's suggested to make clear the clinical significance of these variables with LEAD first before doing the analyses.

7. In the first part of the discussion, it's suggested to make a brief introduction of the main findings first. Strengths and limitation should be clarified in the discussion part also.

8. The key finding is the prediction model in this study. So, please clarify the main findings compared with other studies instead of talking about each risk factor in the model.

9. It's suggested to reorganize the manuscript to highlight the main findings from this study.

Reviewer #4: This paper presents a highly interesting study that offers valuable "big data" insights into a particularly challenging disease. I have the following comments :

1. In Table 1, you present the characteristics of all patients included in the dataset. However, it is unclear who is responsible for collecting this data and by what methods. It is essential to provide more detailed information about the Diabetes Complications Warning Dataset. Specifically, who is responsible for validating the accuracy of the data?

2. Several parameters listed in Table 1 (e.g., arterial hypertension, renal failure, fatty liver, etc.) have definitions established by published standards. How were these parameters defined within your dataset? In the absence of such definitions, how can you ensure consistency in the diagnosis of conditions like renal failure across different patients?

3. Could you elaborate on the rationale behind choosing a 7:3 ratio for the training and validation sets?

4. Your statement regarding atmospheric changes appears to be contentious. The paper you cite associates atmospheric temperature with acute limb ischemia, rather than LEAD. How does your paper’s definition of LEAD compare with the definition used in the study by Chen et al.?

6. PLOS authors have the option to publish the peer review history of their article (what does this mean?). If published, this will include your full peer review and any attached files.

Reviewer #1: No

Reviewer #2: **Yes: **Aram Baram

Reviewer #3: No

Reviewer #4: **Yes: **Theodosios Bisdas

---

## [Author Response · Author response to Decision Letter 0]

12 Sep 2024

PONE-D-24-26075

Risk Factors and Predictive Model Construction for Lower Extremity Arterial Disease in Diabetic Patients

PLOS ONE

Dear Editors and Reviewers,

Thank you for your letter and for the reviewers' comments concerning our manuscript entitled "Risk factors and predictive model construction for lower extremity arterial disease in diabetic patients" (ID: PONE-D-24-26075). The comments provided are valuable and very helpful for revising and improving our paper, as well as having significant guiding implications for our future research. We have carefully considered the comments and made the necessary corrections, which we hope will meet with your approval. The revisions have all been marked in the revised manuscript.

Responding to the journal's requirements:

We sincerely apologize for overlooking PLOS ONE's style requirements. In this revision, we have carefully followed the journal's style guidelines and made the necessary adjustments to our manuscript.

Regarding code sharing, we were unable to access the PLOS Computational Biology author guidance provided by the journal, so the code used in our study was not uploaded in this revision. Our research, which utilizes the Python-based scikit-learn package, focuses on analyzing risk factors for LEAD in diabetic patients and constructing a risk prediction model to help prevent LEAD in this population. This study is more oriented towards medical objectives rather than the improvement of machine learning code. Therefore, we are not currently eager to share our code. However, we would be happy to provide the code upon request.

We update our financial disclosure as follows: "This work was supported by the Shandong Provincial Natural Science Foundation (ZR2022MH268). The funders had no role in study design, data collection and analysis, decision to publish, or preparation of the manuscript."

Regarding data sharing, we will update the data availability statement in this revision to comply the journal's requirements in our manuscript.

Thank you very much for your valuable feedback and assistance in improving our manuscript.

Responds to the reviewer's comments:

Reviewer #1:

1. Response to comment: Treatment factors were excluded in the prediction model. Please clarify what treatment and also provide more detail about the rational for this decision.

Response: 

Thank you very much for your valuable feedback. Most diabetic patients use medications such as insulin to control their blood glucose levels, and there are significant variations in both the types and dosages of these medications. If these patients also suffer from other conditions, the treatment regimen becomes even more complex. Additionally, considering the healthcare environment in China, many patients may have already begun treatment before being admitted, but their prior treatment plans may not have been appropriate. Given these practical conditions, we decided not to include treatment factors in this study design. The main reasons are as follows:

a. Including treatment factors would inevitably make the model overly complex, requiring a sample size beyond our capacity.

b. Collecting data on patients' previous treatment plans would largely rely on the accounts of the patients and their families, who lack a professional medical background. These accounts may contain inaccuracies that could affect the experimental results.

c. We aimed to make this predictive model accessible to non-professionals, so it was important to simplify the model as much as possible without compromising its discriminative ability. This would lower the threshold for use and help more LEAD patients receive treatment earlier.

d. In the future, to enable the automated operation of this predictive model, we need to rely as much as possible on data already available in the medical system, such as diagnoses and laboratory test results. Avoiding the need for secondary data entry would enhance the model's clinical usability.

e. The effects of treatment measures are somewhat reflected in patients' various clinical indicators, so we believe that excluding treatment factors would have an acceptable impact on the model.

For these reasons, we decided not to include treatment factors in this study.

2. Response to comment: Authors only said "missing values were handled" but no further information. How were they handled should be provided.

Response: 

We sincerely apologize for not providing the details on how we handled missing values. For the missing data in our study, we addressed it by removing incomplete cases. We have revised the resubmitted manuscript to include these details. Specifically, in lines 111-115, we changed "Missing values were handled before model training" to "Before training the model, we excluded incomplete cases from the dataset, resulting in the removal of 15 cases. After this preprocessing step, a total of 2,985 cases were included in the model."

3. Response to comment: Also, testing collinearity is good. But what authors deal with collinear variables should be provided.

Response: 

Thank you very much for your feedback. We examined the multicollinearity of the relevant risk factors using the df.corr() method from the pandas library in Python.

4. Response to comment: All significant predictors from univariate analysis were included in multivariate analysis. How about those non-significant in univariate analysis? If not included, will you be concerned about the potential confounding?

Response: 

We apologize for not providing a detailed explanation on this issue. Since the differences observed in the results of univariate analysis may not accurately reflect the effect of each factor on the outcome event, we decided to include variables with statistical significance (P < 0.05) from the univariate analysis into the multivariate analysis. Additionally, to avoid omitting important variables, we also extended the inclusion criteria to P < 0.1. This approach is commonly used in medical research, and many well-regarded studies have employed this method.

Therefore, in our study, we included factors with P < 0.1 from the univariate analysis in the multivariate analysis. To clarify this issue, we have added this explanation to the revised manuscript. In lines 107-110, we changed "All significant predictors identified in the univariate analysis were incorporated into the multivariate analysis" to "Based on the results of the univariate analysis, factors with a p-value less than 0.1 were included in the multivariate analysis." In lines 158-162, we revised "In the multivariate analysis, all variables that showed a significant association with LEADDP in the univariate analysis were included" to "In the multivariable analysis, all variables that showed a strong association with LEADDP in the univariate analysis (P < 0.1) were included."

5. Response to comment: Please present the sample characteristic comparison between the training and testing sample.

Response: 

Thank you very much for your suggestion. In this study, we overlooked providing the sample characteristics for the training and testing samples. After this revision, we have included a detailed description of the sample characteristics in the text. For example, in lines 117-119, we have updated it to: "The training set included 2,089 cases, with 296 cases of LEADDP and 1,793 cases of non-LEADDP. The validation set included 896 cases, with 127 cases of LEADDP and 769 cases of non-LEADDP." To make it easier for readers to understand the sample characteristics of both the training and testing samples, we have created a table that directly presents these characteristics. Please refer to Table 1 for more details.

Table 1. Characteristics of Samples in the Training and Validation Sets

 NO LEADDP LEADDP Total

Training set 1793 296 2089

Validation set 769 127 896

Total 2562 423 2985

The above content constitutes our responses to your comments and the revisions made to the manuscript based on your suggestions. We must express our gratitude to you for your valuable feedback. Your recommendations, which address aspects such as study design, data preprocessing, and statistical analysis, have significantly contributed to improving the readability of our manuscript and enhancing our research. Thank you once again.

Reviewer #2:

1. Response to comment: It needs a minor revision , of the statistical methods with better description of the patients and methods.

Response: 

Thank you very much for your comments. We have revised the description of the statistical methods, further explained the considerations and reasons for including risk factors in our study, supplemented the criteria for including risk factors in the multivariate analysis, and provided details on the handling of missing values. We have also presented the characteristics of the training and testing samples.

In lines 79-82, we added: "Due to the complex pathogenesis of LEADDP and the unclear associated risk factors, we also included more information on comorbidities and additional laboratory indicators in order to identify new risk factors and avoid omissions" to further explain the rationale for including risk factors in this study.

The original Table 1 has been moved to the supplementary materials for easier reading.

We also supplemented the criteria for including risk factors in the multivariate analysis, stating: "Based on the results of the univariate analysis, factors with a p-value less than 0.1 were included in the multivariate analysis" (lines 109-110).

We added details on the handling of missing values, as follows: "Before training the model, we excluded incomplete cases from the dataset, resulting in the removal of 15 cases. After this preprocessing step, a total of 2,985 cases were included in the model. The collinearity of the included factors was examined" (lines 113-115).

We described the sample characteristics of the training and validation sets. You can see in lines 117-119: "The training set included 2,089 cases, with 296 cases of LEADDP and 1,793 cases of non-LEADDP. The validation set included 896 cases, with 127 cases of LEADDP and 769 cases of non-LEADDP (Table 1)." This information can be visually assessed from Table 1.

Table 1. Characteristics of Samples in the Training and Validation Sets

 NO LEADDP LEADDP Total

Training set 1793 296 2089

Validation set 769 127 896

Total 2562 423 2985

2. Response to comment: The results are not well detailed and needs more refinement.

Response: 

We apologize for the lack of detail in the results provided. In this revision, we have addressed your comments and further enhanced the research results. We have added the units for the risk factors included in this study, which were overlooked in the initial draft. Additionally, we have supplemented the 95% confidence intervals for the multivariate analysis results.

In line 130, we added general information about the patients, such as: "The average age was 57.8 years, with 37.5% being female."

In Table 2, we have included the units for each risk factor.

In Table 3, we have refined and completed the results of the multivariate analysis. The revised Table 3 is as follows.

Table 3. Results of the multivariate analysis

 P value OR ( 95% CI )

Sex 0.018 1.422 (1.062, 1.904)

AS ＜0.001 19.911 (12.349, 32.103)

Carotid Artery Stenosis 0.002 2.198 (1.345, 3.591)

FLD 0.004 1.518 (1.140, 2.023）

Hematonosis ＜0.001 2.094 (1.467, 2.989）

Endocrine Disease ＜0.001 2.155 (1.623, 2.862）

GSP 0.014 1.002 (1.000, 1.004）

Abbreviations: AS, Atherosclerosis; FLD, Fatty Liver Disease; GSP, Glycated Serum Protein.

3. Response to comment: The discussions are weakly describing the current literature.

Response: 

Thank you very much for your comments. Following your feedback, we have reorganized the manuscript and revised the discussion section.

At the beginning of the discussion, we added the following content to summarize the findings of our study: "In our study, approximately 15.9% of Chinese diabetic patients had LEAD. Male gender, atherosclerosis, carotid artery stenosis, fatty liver, hematologic diseases, other endocrine disorders, and glycated serum protein were found to be independently associated with the prevalence of LEADDP. The risk prediction model constructed using Logistic regression and MLP algorithms achieved the best performance, with moderate discriminative ability" (lines 174-179).

To more accurately explain the differences between our study and a previous study, we included additional details about the study population's regional characteristics. In lines 180-182, we modified "In our study, approximately 15.9% of Chinese diabetic patients had LEADDP, whereas another study reported a prevalence of about 4.9%" to "In our study, approximately 15.9% of Chinese diabetic patients had LEADDP, whereas another study from the southern coastal region of China reported a prevalence of about 4.9%." To explain the differences in prevalence between the two studies, we added the following content in lines 183-187: "The reason for this difference is not yet clear but may be related to regional variations. Our study data are from a medical center in northern China, where the climate is colder compared to the south. The diagnosis of LEAD primarily relies on ABI (Ankle-Brachial Index). At low temperatures, the pressure in the distal arteries of the lower limbs significantly decreases, which reduces the ABI value [11]." Additionally, in lines 189-194, we added: "Furthermore, there are significant differences in dietary habits, lifestyles, and economic levels among people in different provinces of China, which may, to some extent, affect the development of diabetes and its complications. Existing research data indicate that the prevalence of diabetes varies across different provinces in China [12]. However, there have been no epidemiological reports specifically on LEADDP in China." These modifications help clarify the differences in prevalence between our study and the previous research on LEAD among Chinese diabetic patients.

In lines 199-201, we revised "However, in our study, we found that the incidence of LEADDP was higher in males" to "However, in our study, we found that there were more men than women among diabetic patients with concurrent LEAD." In line 202, we added the following content and reference: "A study on LEAD in the diabetic population also reached similar conclusions [16]." These changes help explain the gender differences in LEAD prevalence among diabetic versus non-diabetic patients.

In lines 216-218, we added: "However, in our study, there were no significant differences in the prevalence of complications such as coronary heart disease, myocardial infarction, and chronic heart failure," to further discuss our results.

To explain the mechanism by which hyperglycemia damages blood vessels, we added in lines 224-226: "Hyperglycemia promotes oxidative stress, glycoxidation, and systemic inflammation, damaging the endothelial cells of the arterial wall, leading to lipid deposition and the development of atherosclerosis [21]." To clarify the relationship between blood glucose control and LEAD risk in our study, we included in lines 231-234: "In our study, LEAD patients had poorer blood glucose control compared to those without LEAD. Although fasting blood glucose levels did not show significant differences, the postprandial 2-hour blood glucose levels were significantly elevated." To explain our findings, we added in lines 236-238: "Studies have also shown that arterial vascular damage can begin before blood glucose levels increase, with elevated insulin levels potentially being a contributing factor [21]."

To clarify our results regarding lipids, fatty liver, and BMI, we added in lines 252-254: "Elevated low-density lipoprotein levels indicate dyslipidemia. In our study, patients with both LEAD and fatty liver had an increased risk of LEAD, and t

---

## [Decision Letter · Decision Letter 1]

14 Oct 2024

PONE-D-24-26075R1Risk Factors and Predictive Model Construction for Lower Extremity Arterial Disease in Diabetic PatientsPLOS ONE

Dear Dr. Kuang,

Thank you for submitting your manuscript to PLOS ONE. After careful consideration, we feel that it has merit but does not fully meet PLOS ONE’s publication criteria as it currently stands. Therefore, we invite you to submit a revised version of the manuscript that addresses the points raised during the review process.

**Please address comments from reviewer 3. **

We look forward to receiving your revised manuscript.

Kind regards,

Jincheng Wang

Academic Editor

PLOS ONE

**Journal Requirements:**

**Additional Editor Comments :**

Please address comments from reviewer 3.

Reviewers' comments:

Reviewer's Responses to Questions

**Comments to the Author**

1. If the authors have adequately addressed your comments raised in a previous round of review and you feel that this manuscript is now acceptable for publication, you may indicate that here to bypass the “Comments to the Author” section, enter your conflict of interest statement in the “Confidential to Editor” section, and submit your "Accept" recommendation.

Reviewer #1: All comments have been addressed

Reviewer #2: All comments have been addressed

Reviewer #3: (No Response)

2. Is the manuscript technically sound, and do the data support the conclusions?

Reviewer #1: (No Response)

Reviewer #2: Yes

Reviewer #3: Yes

3. Has the statistical analysis been performed appropriately and rigorously? 

Reviewer #1: (No Response)

Reviewer #2: Yes

Reviewer #3: Yes

4. Have the authors made all data underlying the findings in their manuscript fully available?

Reviewer #1: (No Response)

Reviewer #2: Yes

Reviewer #3: Yes

5. Is the manuscript presented in an intelligible fashion and written in standard English?

Reviewer #1: (No Response)

Reviewer #2: Yes

Reviewer #3: No

6. Review Comments to the Author

**Reviewer #1: **(No Response)

**Reviewer #2: **All points are revised; it's a benefit for the specialists in this field, while needs more syntax refinements

**Reviewer #3:** The authors have addressed most of the comments left. However, there still exist some main concerns here. The authors stated that they opted to include more influencing factors in their analysis to identify risk factors associated with LEAD and avoid omissions by using machine learning algorithms. I agree with the authors on this view, but the following points still need to be revised.

First, this study found that “Male gender, atherosclerosis, carotid artery stenosis, fatty liver, hematologic diseases, other endocrine disorders, and glycated serum protein were found to be independently associated with the prevalence of LEADDP.” It makes little sense to include so many comorbidities in developing the risk prediction model for LEAD. Due to the nature of the cross-sectional design, it cannot be determined whether LEAD cause these diseases or vice versa. At first, it’s okay to include various variables as the candidate for risk factor selection. However, the clinical significance of these variables to be selected as variables should be made clear.

Second, the authors said that there existed gender differences in LEAD distribution and the associations of hematologic diseases with LEAD as new findings. “Currently, no studies have analyzed the correlation between hematologic diseases and LEADDP. This finding opens new avenues for research into LEADDP.” So, is there any clinical significance here to study the association between hematologic diseases and LEAD in people with diabetes? And, what’s the clinical or public health significance here to use this model among those with diabetes? For the discussion part, it’s wordy to present discussion with each risk factor found in this study as a single paragraph. It makes no highlight or focus. And, the key problem is still the finding of this study. So many comorbidities with just gender and glycated serum protein as a risk prediction model makes little clinical or public health significance. Or, the focus of the paper is to study the association between cormobidites with LEAD among those with diabetes?

Besides, is there other risk prediction model not using machine learning algorithms? The authors should discuss this in the discussion part. Is there any added value here in this study by using machine learning algorithms? Or, is this study the first one to study the risk prediction model of LEAD in those with diabetes?

7. PLOS authors have the option to publish the peer review history of their article (what does this mean?). If published, this will include your full peer review and any attached files.

Reviewer #1: No

Reviewer #2: **Yes: **Aram Baram

Reviewer #3: No

---

## [Author Response · Author response to Decision Letter 1]

17 Oct 2024

Dear Editors and Reviewers,

Thank you for your letter and for the reviewers' comments concerning our manuscript entitled "Risk factors and predictive model construction for lower extremity arterial disease in diabetic patients" (ID: PONE-D-24-26075). We have carefully considered these comments, and below are our responses.

Responding to the journal's requirements:

Thank you for your reminder; we have reviewed the reference list again.

Responds to the reviewer's comments:

Reviewer #3:

Thank you very much for your comments. Your feedback is thorough and rigorous, which has been very helpful to us. In response to your remarks, we would like to address the following points.

Significance and Public Health Value of the Study

This study aims to predict the risk of LEAD in diabetic patients and to improve the diagnostic rate of LEAD within the diabetic population, facilitating early detection and intervention to prevent the further progression of LEAD.

As stated in our manuscript, "Statistics indicate that over 60% of LEAD patients may be asymptomatic, leading to early stages of the disease often being overlooked. Compared to non-diabetic LEAD patients, diabetic peripheral neuropathy may mask LEAD symptoms, making the diagnosis of LEADDP more challenging. Current guidelines recommend initial screening for LEAD based on patient interviews and clinical examinations, using the Fontaine or Rutherford scales for assessment. This diagnosis heavily relies on the expertise of specialists and examination conditions, and the diagnostic performance is not ideal." (line 49-56)

Due to a lack of specialized knowledge, diabetic patients may not pay attention to symptoms such as reduced skin temperature in the lower extremities or mild pain, leading them to conceal their medical history and miss timely specialist examinations. They might seek care for diabetes or its complications in endocrinology or other departments, rather than choosing vascular surgery or peripheral vascular disease specialties to address LEAD. This could further result in LEAD being overlooked, delaying treatment. Therefore, we aim to address this issue through our study.

Our model predicts the risk of LEAD at a specific moment by integrating the clinical data of patients, allowing for the identification of high-risk populations. To illustrate this point further, we would like to provide an example. Risk assessment for venous thromboembolism (VTE) has been widely implemented in clinical practice, where risk prediction incorporates not only laboratory test indicators but also factors such as comorbidities. While these comorbidities are associated with VTE, this does not imply that they cause VTE to occur. By evaluating the risk of VTE, high-risk patients can be identified early, enabling timely treatment to prevent further progression of the condition. Similarly, our study aims to identify high-risk LEAD patients to facilitate early detection and intervention.

In our research, we have incorporated machine learning algorithms. Compared to traditional risk prediction methods, machine learning algorithms are better equipped to handle complex datasets containing multiple variables and features. More importantly, the risk prediction models constructed using machine learning have the potential for full automation, thereby reducing the need for healthcare personnel intervention, lowering clinical workloads, and improving diagnostic efficiency. This prospect holds significant social value.

Therefore, we believe that this study holds significant clinical importance and public health value.

Relationship Between Risk Factors and LEAD

LEAD is part of a broader spectrum of vascular diseases, with complex pathophysiological mechanisms. Although numerous studies exist, the detailed pathogenic mechanisms remain unclear. In the diabetic population, LEAD may have unique mechanisms of onset. While these pathogenic mechanisms are causally related to LEAD, their ambiguity prevents us from incorporating them into our risk prediction for LEAD.

By analyzing the relationship between various risk factors and LEAD, we can identify comorbidities that share similar pathogenic mechanisms with LEAD. In our study, we found an association between hematologic diseases and LEAD. Although the common mechanisms underlying this relationship remain unclear, this finding may provide insights for future research. To illustrate this point, we can use the example of patients with polycythemia vera, who may simultaneously experience headaches and lower limb venous thrombosis. While there is an association between headaches and venous thrombosis, this does not imply a causal relationship. By analyzing these associations and conducting further research, we may uncover changes in the patients' blood cells and ultimately identify the mutated genes.

Such findings can help us better understand the pathophysiological processes of LEAD, laying the groundwork for developing more effective medical interventions. Considering these factors, we intend to dedicate a section of our discussion to exploring the potential associations between various risk factors and LEAD in the diabetic population.

Comparison of This Study with Similar Research

You pointed out that "the discussion section should include relevant studies on LEAD risk prediction models," and we agree with your observation. This supplementary section will be outlined in the following paragraph. While we believe that more attention is needed for research on LEAD in diabetic patients, it is important to note that some studies on LEAD risk prediction models already exist. Some of these studies have employed machine learning algorithms, while others have not. Compared to studies that did not use machine learning algorithms, the introduction of machine learning can effectively handle a greater number of variables and complex features, yielding better performance. Furthermore, by continuously updating data and retraining the models, the predictive capabilities can be further enhanced. Additionally, improving machine learning algorithms to establish diagnostic platforms can lower the barrier to use and enhance diagnostic efficiency, offering broader application prospects. Compared to similar studies that utilized machine learning algorithms, the risk prediction model constructed in this study demonstrates certain performance advantages. Moreover, to our knowledge, this is the first study to develop a LEAD risk prediction model using machine learning algorithms specifically for the diabetic population in China.

In response to this issue, we have made additions in the revised manuscript. In lines 258-263, we included the following content: "Previous studies have employed machine learning algorithms to develop risk prediction models for LEAD, achieving relatively good predictive performance. However, these studies did not specifically focus on diabetic patients, and thus could not capture the unique characteristics of LEAD occurrence in this population. The application of machine learning algorithms in LEAD risk prediction models specifically targeting diabetic patients remains relatively limited."

Other changes:

We have added two references in lines 437-444.

Finally, we would like to express our gratitude once again for your assistance in improving this study.

---

## [Decision Letter · Decision Letter 2]

19 Nov 2024

Risk Factors and Predictive Model Construction for Lower Extremity Arterial Disease in Diabetic Patients

PONE-D-24-26075R2

Dear Dr. Kuang,

We’re pleased to inform you that your manuscript has been judged scientifically suitable for publication and will be formally accepted for publication once it meets all outstanding technical requirements.

Kind regards,

Jincheng Wang

Academic Editor

PLOS ONE

Additional Editor Comments (optional):

Authors have addressed all comments.

Reviewers' comments:

Reviewer's Responses to Questions

**Comments to the Author**

1. If the authors have adequately addressed your comments raised in a previous round of review and you feel that this manuscript is now acceptable for publication, you may indicate that here to bypass the “Comments to the Author” section, enter your conflict of interest statement in the “Confidential to Editor” section, and submit your "Accept" recommendation.

Reviewer #3: All comments have been addressed

2. Is the manuscript technically sound, and do the data support the conclusions?

Reviewer #3: Yes

3. Has the statistical analysis been performed appropriately and rigorously? 

Reviewer #3: Yes

4. Have the authors made all data underlying the findings in their manuscript fully available?

Reviewer #3: Yes

5. Is the manuscript presented in an intelligible fashion and written in standard English?

Reviewer #3: Yes

6. Review Comments to the Author

Reviewer #3: The authors of this manuscript have addressed all the comments. No further questions to the authors.

7. PLOS authors have the option to publish the peer review history of their article (what does this mean?). If published, this will include your full peer review and any attached files.

Reviewer #3: No

---

## [Editor Report · Acceptance letter]

17 Dec 2024

PONE-D-24-26075R2 

PLOS ONE

Dear Dr. Kuang, 

I'm pleased to inform you that your manuscript has been deemed suitable for publication in PLOS ONE. Congratulations! Your manuscript is now being handed over to our production team.

Kind regards, 

on behalf of

Dr. Jincheng Wang 

Academic Editor

PLOS ONE